**RESEARCH**

# A benchmark of computational pipelines for single-cell histone modification data

Félix Raimundo[1,2], Pacôme Prompsy[2,3], Jean-Philippe Vert[1,4*] and Céline Vallot[2,3*]

*Correspondence:
jean-philippe.vert@owkin.com;
celine.vallot@curie.fr

[1] Google Research, Brain team, 75009 Paris, France
[2] Translational Research Department, Institut Curie, PSL Research University, 75005 Paris, France
[3] CNRS UMR3244, Institut Curie, PSL Research University, 75005 Paris, France
[4] Owkin, Inc, NY, New York, USA

## Abstract

**Background:** Single-cell histone post translational modification (scHPTM) assays such as scCUT&Tag or scChIP-seq allow single-cell mapping of diverse epigenomic landscapes within complex tissues and are likely to unlock our understanding of various mechanisms involved in development or diseases. Running scHTPM experiments and analyzing the data produced remains challenging since few consensus guidelines currently exist regarding good practices for experimental design and data analysis pipelines.

**Results:** We perform a computational benchmark to assess the impact of experimental parameters and data analysis pipelines on the ability of the cell representation to recapitulate known biological similarities. We run more than ten thousand experiments to systematically study the impact of coverage and number of cells, of the count matrix construction method, of feature selection and normalization, and of the dimension reduction algorithm used. This allows us to identify key experimental parameters and computational choices to obtain a good representation of single-cell HPTM data. We show in particular that the count matrix construction step has a strong influence on the quality of the representation and that using fixed-size bin counts outperforms annotation-based binning. Dimension reduction methods based on latent semantic indexing outperform others, and feature selection is detrimental, while keeping only high-quality cells has little influence on the final representation as long as enough cells are analyzed.

**Conclusions:** This benchmark provides a comprehensive study on how experimental parameters and computational choices affect the representation of single-cell HPTM data. We propose a series of recommendations regarding matrix construction, feature and cell selection, and dimensionality reduction algorithms.

## Background

Posttranslational modifications (PTM) of histone proteins are key epigenetic events that modulate chromatin structure, nucleosome positioning and transcription. They are involved in numerous biological processes, including DNA repair [1], development [2, 3], and cancer [4]. With the recent advent of high-throughput technologies to measure

histone PTM at the single-cell level (scHPTM), such as single-cell chromatin immuno-precipitation followed by sequencing (scChIP-seq) [5] and single-cell cleavage under targets and tagmentation (scCUT &Tag) [6, 7], it is now feasible to explore the diversity of histone PTM in complex biological samples with an ever-increasing level of details [6, 8–10]. ScHPTM has already allowed new biological insights such as the discovery of epigenetic factors involved in cancer response to chemotherapy [11] and is likely to unlock our understanding of various epigenetic mechanisms in the years to come.

While scHPTM has great potential, it is also a relatively recent approach which comes with numerous computational challenges that need to be addressed in order to fully deliver its promise of capturing biologically relevant information from raw experimental data. In this work, we leave aside the question of which technology to use to generate scHPTM data and focus instead on two important questions for practitioners, namely, (1) how to design experiments, in particular to choose a good trade-off between number of cells and coverage, and (2) how to computationally analyze the raw experimental data and transform them in biologically relevant representations, where subsequent analysis such as cell classification or lineage inference become feasible. While both questions have been investigated through systematic benchmarks and comparisons for more mature single-cell technologies such as single-cell RNA-seq (scRNA-seq) and single-cell sequencing assay for transposase-accessible chromatin (scATAC-seq) [12–16], we are not aware of any similar study conducted for the burgeoning field of scHPTM, leaving experimentalists without rational guidelines on how to design their scHPTM experiments and analyze the data they produce.

Given the similar nature of raw experimental data between scHPTM and scATAC-seq, namely, sequencing reads capturing an epigenomic signal distributed in specific regions over the whole genome, it would seem natural to use the same computational methods to analyze scHPTM and scATAC-seq data. However, both modalities differ in many aspects. First, the actual distribution of reads can be drastically different between scHPTM and ATAC-seq. Indeed, ATAC-seq reads are known to cluster in relatively small, ~1k base pairs (kbp), regions [17], whereas the regulatory regions for scHPTM vary much more widely in size (e.g., between 5 kbp and 2000 kbp for H3K27me3 [17]), and their locations can vary depending on the histone mark——from enhancers (H3K27ac) to gene body (H3K36me3) or intergenic regions (H3K27me3). Second, with current technologies, the number of sequenced reads in scHPTM is generally between a few hundred and a few thousand per cells, compared to several thousands for scATAC-seq and tens of thousands for scRNA-seq. Such a low coverage leads to only about 1% of the expected enriched regions to contain at least one read per cell (compared to 1–10% for scATAC-seq and 10–45% for scRNA-seq [12]). Thus, one can not assume that computational recommendations for scATAC-seq or RNA-seq hold for scHPTM.

To start filling this gap, we perform in this paper a large-scale computational study to evaluate the impact on embeddings and performances of different parameters: (i) the number of cells, (ii) coverage per cell, (iii) cell selection, (iv) matrix construction algorithm, (v) feature selection, and (vi) dimension reduction algorithm. The analysis of more than 10,000 computational experiments allows us to clarify the impact of various experimental choices and data processing factors for scHTPM data and to suggest practical guidelines. To quantify the impact of each of these factors, we use neighbor

scores——based on the comparison of epigenomic and transcriptomic embeddings when possible——as well as Adjusted Mutual Information (AMI) or Adjusted Rand Index (ARI) to compare to reference labels. For two single-cell multi-omics datasets, in addition to scHPTM, a second modality is measured for each cell (gene expression or cell surface proteins); in this case, we can assess neighbor score, i.e., how well the cell-to-cell similarity observed with scHPTM data analysis agrees with the one inferred from the co-assay (RNA or protein) [18, 19].

## Results

### Benchmarking methods for scHPTM analysis

Irrespective of the technology used, most protocols for scHPTM analysis produce sequencing reads which, after being mapped to a reference genome, indicate where on the genome a given PTM mark is likely to be present in each individual cell under study. A number of computational steps are then applied to transform these raw data into a useful representation of each individual cell, where downstream applications such as cell classification or differential analysis are performed. Here, we focus on computational frameworks that produce a representation of each cell as a vector of moderate dimension (typically, 10 to 50 dimensions), which has been found to be a powerful approach for scRNA-seq data analysis [20] and is currently the de facto standard for scATAC-seq and scHPTM as well [20]. Going from the mapped read to a vector representation for each cell involves a number a steps that we investigate in this study (Fig. 1A), including (1) the binning of the mapped reads into genomic regions in order to create a cell×region count matrix to summarize the raw data, (2) various quality control (QC) preprocessing operations to filter out low-quality cells and regions, and (3) an embedding method to build the representation of each cell from the preprocessed count matrix. Each step can be performed in many different ways, and we propose a benchmark to assess the impact of each choice at each step on the final cell representation (Fig. 1).

In order to evaluate the impact of each decision on the quality of the final representation, we need reference datasets and a way to quantify the quality of that representation. A standard approach to measure this is to evaluate the performances of clustering algorithms on these representations (such as in [13]). This approach requires cell type annotations, which can either be computationally derived or experimental (e.g., through cell lines or FACS). scHPTM protocols being rather recent, datasets with experimental labels are rare; here, we use the dataset from [11], with treated or untreated breast cancer cells as experimental labels. In addition, due to this lack of sufficient ground truth reference datasets for scHPTM analysis, we also rely on two datasets produced with multiomics co-assays (Table 1), where two modalities are measured simultaneously in each cell. More precisely, we consider a mouse brain dataset from [9] where five histone marks (H3K4m1, H3K4me3, H3K9me3, H3K27ac, and H3K27me3) are assessed by scHPTM jointly with scRNA-seq-based gene expression, and a human peripheral blood mononuclear cell (PBMC) dataset from [10] where the same five histone marks are assessed by scHPTM jointly with CITE-seq-based cell surface proteins. For both datasets, we use a unique representation of the second modality (respectively, scRNA-seq and CITE-seq) using a well-established method as a reference (scanpy's [21] implementation of PCA) and compare each representation obtained from the scHPTM data

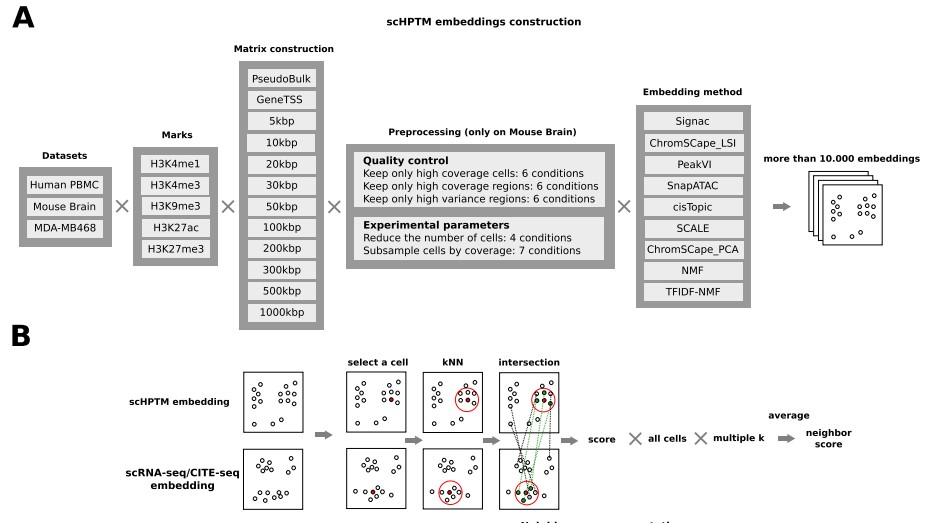

**Fig. 1** Overview of the evaluation protocol. **A** We build the count matrix using different bin sizes as well as a GeneTSS annotation and peaks called on the pseudo bulk with both MACS2 and SICER (only for the human PBMC dataset), for the MDA-MB468 dataset we also used author provided annotations. We then simulate in silico different experimental conditions for studying the role of the number of cells in a dataset, and the effect of the coverage per cell, as well as different feature selection strategies. Afterwards, we run 9 different dimension reduction methods to obtain the cell representations. **B** In order to compute the neighbor score, we start by selecting a cell, we then build the *k*NN graph for a value of *k* (5 in the figure), and we then compute the size of the intersection between the neighborhood of the cell in the two embeddings (3 cells in the figure) and divide it by *k* to obtain the score for one cell and one value of *k* (score of 0.6 in the figure). We then compute and average this score over all the cells, to have an neighbor score for a given value of *k*; that score is then further averaged over different values of *k* (0.1%, 0.3%, 0.5%, 1%, 3%, 5%, and 10% of the number of cells in the experience) to obtain the final neighbor score

**Table 1** Description of the datasets used for this study

| Tissue | Source | Co-assay | Mark | Number of cells |
|---|---|---|---|---|
| Mouse brain | [9] | RNA-seq | H3K4me1 | 12,962 |
| | | | H3K4me3 | 7465 |
| | | | H3K9me3 | 12,044 |
| | | | H3K27ac | 11,749 |
| | | | H3K27me3 | 6534 |
| Human PBMC | [10] | CITE-seq | H3K4me1 | 12,770 |
| | | | H3K4me3 | 10,386 |
| | | | H3K9me3 | 8304 |
| | | | H3K27ac | 15,609 |
| | | | H3K27me3 | 8232 |
| MDA-MB468 | [11] | None | H3K27me3 | 6031–9840 |

to that reference. We compute a neighbor score that assesses to what extent neighbor cells in the scHPTM representation are also found neighbors in the reference representation of the second modality. The neighbor score varies between 0 when both representations disagree completely to 1 when both representations are identical (see Methods and Fig. 1B). This evaluation has been previously used in [18, 19] and is currently the standard for evaluating modality alignment tasks in recent community benchmarks such

as https://openproblems.bio/. We furthermore use the initial labels from authors——computationally derived——to evaluate representations with ARI and AMI.

For each dataset and each histone PTM mark, we systematically vary the choices that we can make in each step of the computational pipeline that goes from the mapped reads to the scHPTM representation of each cell and measure the quality of the final representation with the neighbor score to assess the impact of the choices.

More precisely, for the first step that bins mapped reads to regions in order to build a first cell×region count matrix, we consider three different strategies that represent the various approaches used in practice for the analysis of single-cell epigenomic assays: (1) discretizing the whole genome into "bins" of fixed size and trying different sizes following a logarithmic progression between 5 kbp and 1000 kbp, (2) counting the reads into bins based on prior biological knowledge, i.e., on genes and transcription start sites annotations (GeneTSS), and (3) counting the reads into a set of peaks, characteristic of each cell population found in the sample (identified from the corresponding pseudo bulk using MACS2 [22], "MacsPseudoBulk," or SICER [23], "SicerPSeudoBulk"). This last approach was only performed with the human PBMC dataset that is distributed in a format that allows us to build the pseudo bulk and use it for peak calling. With these matrices, we attempt different feature selection approaches to select only a subset of genomic regions to keep for further analysis: (1) selection of highly variable regions using Seurat's [24] FindVariableFeatures function (variable features) and (2) selection of regions with the highest coverage (top features). The first feature selection method is the current standard in scRNA-seq, and the second approach is recommended in Signac [25] for analyzing scATAC-seq. We further study the role of cell filtering based on their coverage, which is part of the standard analysis steps. For region filtering, we study the effect of coverage and variance filtering. We also simulate different experimental conditions in silico in order to evaluate how cell numbers affect cell representation, as well as the importance of their coverage. Finally, we consider nine popular methods for analyzing the count matrices: cisTopic [26], Signac [25] SnapATAC [27], PeakVI [18], SCALE [28], ChromSCape [29] with TF-IDF (ChromSCape_LSI) and count per million (CPM) normalization (ChromSCape_PCA), and NMF with no normalization (NMF) or TF-IDF transformation (TFIDF-NMF).

This leads us to test 11,970 combinations of mark, dimension reduction method, matrix construction, cell selection, feature selection, number of cells, and coverage conditions, out of which 11,080 ran successfully (Additional file 1: Tables S1-S2). Failures to run were generally due to memory issues on small bin sizes and GeneTSS annotation. We then analyze the impact of each decision choice and experimental condition by assessing statistically how the neighbor score of the representation varies with the decision.

### TF-IDF-based methods outperform other methods

We first focus on the influence of the embedding methods on the quality of the final representation. The nine methods we selected implement a broad range of algorithms that are currently used for the analysis of scATAC-seq and scHPTM data. More precisely, ChromSCape_PCA is a simple use of PCA after count per million (CPM) normalization, which serves as baseline. ChromSCape_LSI and Signac implement two variants of

the latent semantic indexing (LSI) algorithm, which consists in transforming the count matrix with TF-IDF and applying PCA on that matrix. They have been used to analyze scHPTM data [10, 29], and differ in the fact that ChromSCape_LSI weights the principal components by their eigenvalues, as is standard to do with PCA, while Signac does not and instead whitens the data representation. They implement variants of the algorithm used in Cusanovich2018 [30–32], which was found with SnapATAC and cisTopic to be among the best methods for scATAC-seq data analysis in [12]. NMF has also been used in combination to TF-IDF to analyze single-cell datasets, in scOpen [33]. SnapATAC computes the Jaccard similarity between all the cells, and runs kernel PCA on this similarity matrix. cisTopic binarizes the count matrix and then applied latent Dirichlet allocation (LDA) on this modified matrix. Finally, SCALE and PeakVI both implement a variational autoencoder (VAE) with a product of Bernoulli likelihood function. They differ in the fact that SCALE uses a mixture of gaussian prior where PeakVI uses a unimodal gaussian prior. Furthermore, PeakVI computes corrections for the size factor of each cell as well as for the accessibility of each DNA region. We run all methods with their default parameters (see Methods). In particular, we keep the default number of dimensions for all methods; indeed, some methods offer their own heuristics for deciding the number of dimensions, and we did not want to disadvantage them by using a dimension they do not consider optimal. More precisely, PeakVI sets by default the dimension to the square root of the square root of the number of regions, while cisTopic trains model for multiple dimensions and chooses one based on an elbow rule of its evidence lower bound (ELBO). Signac uses a dimension of 50 by default, while SnapATAC, SCALE, NMF, TFIDF-NMF, and ChromSCape have a default dimension of 10.

Figures 2A and Additional file 1: Fig. S4 summarize the performance of each embedding method on the different histone PTM marks in the mouse brain and human PBMC datasets, respectively. In those plots, we summarize the performance of each embedding method by reporting the best performance achieved by each embedding method across all possible matrix construction choices, without performing any additional QC processing such as cell or feature selection. This allows us to quantify the best possible result that each embedding method can reach without setting an arbitrary feature engineering pipeline that could advantage some methods over others. We see that the neighbor scores vary roughly in the range 0.05~0.35 across methods, datasets, and marks. As can be seen in Fig. 2B, where we visualize the embeddings obtained by ChromSCape_LSI on different marks on the mouse brain dataset, this corresponds to a fairly good agreement with scRNA-seq embedding in terms of recovering major cell types, particulary for H3K27ac (score = 0.302) and H3K4me1 (score = 0.321). Interestingly, we observe differences in the neighbor scores of different marks across methods in the mouse brain dataset, with H3K4me1 and H3K27ac (score = 0.291 ± 0.028 and 0.273 ± 0.026, respectively) generally higher than H3K9me3 and H3K27me3 and H3K4me3 (score = 0.148 ± 0.040, 0.169 ± 0.033, and 0.112 ± 0.035, respectively). Note that this does not necessarily mean that some marks are more informative than others but rather than they are less directly linked to expression than others. A similar trend is visible but weaker on the human PBMC dataset (Additional file 1: Fig. S4), where in particular the scores on H3K27ac and H3K4me1 are lower than on the mouse brain dataset (scores=0.113 ± 0.031 and 0.150 ± 0.021, respectively). This difference between the mouse brain and human PBMC

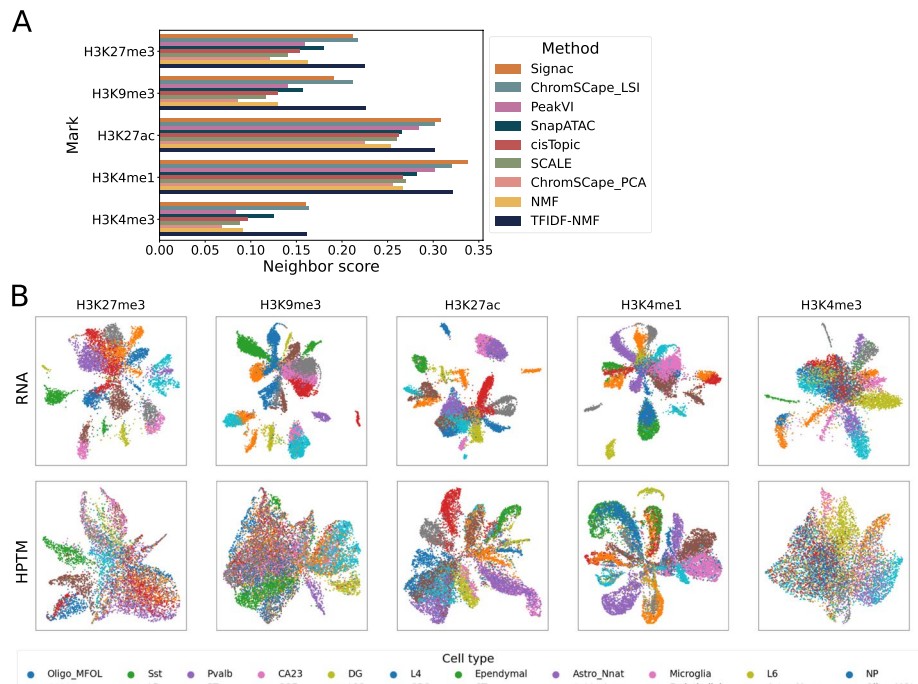

**Fig. 2 A** Best performances of the different representation methods on the mouse brain dataset. **B** UMAP representation of the different samples in the mouse brain dataset; the first row is the RNA co-assay processed with PCA using the scanpy best practices; the second row is the scHPTM assay processed with ChromSCape_LSI using the matrix construction algorithm with the best neighbor score, both colored by the labels of [9] obtained from the scRNA-seq co-assays

datasets could be caused by the differences in co-assay, by the relative complexity of the cell types, or by the quality of the experiments.

The performance of each method on each histone PTM mark of the mouse brain datasets is shown in Fig. 2 and Additional file 1: Table S3. We see that the three best performing methods on the mouse brain datasets are consistently ChromSCape_LSI , TFIDF-NMF, and Signac, which are consistently better than all other methods. They are followed by SnapATAC and PeakVI (except on H3K4me3), then cisTopic, NMF, SCALE, and ChromSCape_PCA. SnapATAC is better than cisTopic and SCALE, while ChromSCape_PCA is worse than all other methods. The top three performing methods (ChromSCape_LSI , TFIDF-NMF, and Signac) implement TF-IDF, suggesting that TF-IDF-based method have an advantage over other approaches. Surprisingly, though, while ChromSCape_LSI and TFIDF-NMF also performs well on the human PBMC dataset, Signac does not (Additional file 1: Fig. S4). This may be due to the lower coverage of the human PBMC dataset than of the mouse brain data, and to the detrimental effect of the whitening operation specific to Signac, as studied in more details in the supplementary text. On the PBMC dataset, ChromSCape_PCA again performs poorly compared to other methods, while the differences between other methods and between marks are overall less pronounced than on the mouse brain dataset.

Since the four methods ChromSCape_PCA, ChromSCape_LSI, Signac, and SnapA-TAC all implement a form of PCA after applying to the count data matrix a specific data transformation, the difference in their performance highlights the importance of this

data transformation choice. Simply normalizing the counts by CPM, as ChromSCape_PCA does, leads to poor performances, while normalizing the count data by Jaccard similarity (SnapATAC) or TF-IDF (ChromSCape_LSI and Signac) is consistently better. This effect of TF-IDF transformation can also be observed in the difference of performances between NMF and TFIDF-NMF. This seems to be specific to scHTPM, since methods using CPM normalization are competitive with the ones using TF-IDF or kernel PCA on the Jaccard similarity on scATAC-seq data [12].

We also find that cisTopic is not among the best performing methods for the analysis of scHPTM, while it was identified by [12] as one of the best tools for analyzing scATAC-seq. On the other hand, LSI is extremely competitive for both modalities. This shows that while scHPTM and scATAC-seq have some similarities, one should be careful before extrapolating good practices from one modality to the other. Finally, the more recent VAE-based methods, PeakVI and SCALE, are overall not competitive with the more classical LSI-based ones. As we show below, this may be due to the relatively small size of the datasets used.

Using the labels provided by the authors in their papers, obtained computationally on the co-assay, we can also compute supervised evaluations of these representations. By using a clustering algorithm, either *k*-means or hierarchical clustering (HC) (Additional file 1: Fig. S7), we obtain clusters that we can compare to the cell annotations provided by the authors using either the Adjusted Mutual Information (AMI) or Adjusted Rand Index (ARI). AMI and ARI are extremely similar as can be seen in Additional file 1: Fig. S6.. The ARI for the mouse brain, human PBMC, and human cell lines datasets are presented in Additional file 1: Fig. S8-S10 respectively. According to this metrics, we observe that the best performing methods for Tn5-based methods are still the three TF-IDF-based methods, with PeakVI also becoming competitive. For the MNase-based method, TFIDF-NMF is still the best performing method, but all methods perform well, except for Signac. This might be due to the low complexity of this dataset, and the default use of 50 whitened dimensions, that can introduce noise in the representation.

### The count matrix construction strongly influences the quality of the representation

We now investigate the influence of the count matrix construction method (i.e., how the raw reads are mapped to regions) to obtain relevant embeddings of scHTPM datasets. For that purpose, we explore the performance of the different embedding methods as a function of the matrix construction parameter, again without further preprocessing such as cell or feature selection. We show the results in Fig. 3 and Additional file 1: Fig, S5 for the mouse brain and human PBMC datasets, respectively.

We see that matrix construction has overall a strong influence on the quality of the representations. For most methods and marks, the performance first increases when the bin size increases, then decrease after a peak. This effect is more pronounced on the mouse brain data and in particular for repressive marks (H3K27me3 and H3K9me3, 3). We also observe for repressive marks studied by scChIP-seq the same increase with bin size as can be seen in Additional file 1: Fig, S10B. In order to quantify this effect, we report the ratios between the best and worst performing matrix construction for each method and mark in Additional file 1: Table S6 for the mouse brain dataset and in Table S7 for the human PBMC dataset. In the human PBMC dataset, we can see that

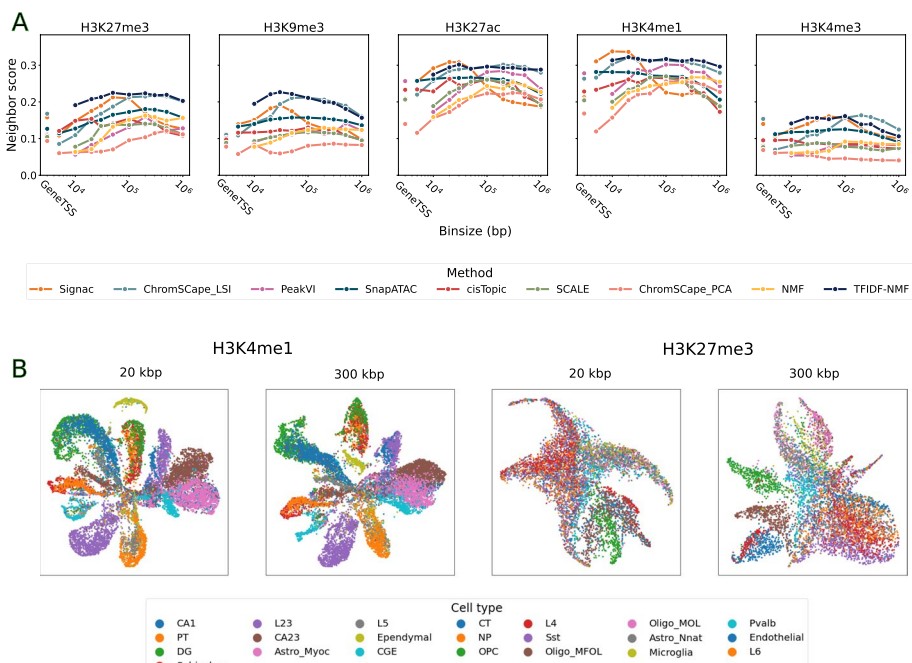

**Fig. 3** **A** Neighbor score performances of the 9 dimension reduction algorithms on the 5 marks in the mouse brain dataset, as a function of the matrix construction. **B** UMAP projection of H3K4me1 and H3K27me3 using ChromsSCape_LSI using bins of 20 kbp and 300 kbp, colored by the labels of [9] obtained from the scRNA-seq co-assays

the ratio between the best and worst feature engineering can reach up to 10.76 (TFIDF-NMF on H3K4me1); this is mostly due to the very poor performances of using a GeneTSS annotation on this dataset as can be seen in Additional file 1: Fig. S5.

In the mouse brain dataset, we can see that this ratio is on average 1.98 for H3K27me3 in Additional file 1: Table S5 and reaches 2.8 in the case of PeakVI. The lowest ratio is 1.09 (TFIDF-NMF on H3K4me1), which is still an increase in performance of 9%. While that ratio is on average higher for the best performing methods (ChromSCape_LSI and Signac), it is mostly due to the fact that their best performances are higher than the other methods, more than it is due to an extreme sensitivity to matrix construction. Indeed, we can see that for all marks, ChromSCape_LSI has a very large range of matrix construction parameters that are extremely competitive. We can also note that by choosing an average performing method (e.g., SnapATAC or PeakVI) and an appropriate matrix construction parameter, we can often beat the best performing methods (TFIDF-NMF, ChromSCape_LSI, or Signac) if they are run with a suboptimal parameter for matrix construction.

We see on the mouse brain dataset that performances reach a level close to their maximum for smaller bin sizes for enhancing marks (H3K27ac, H3K4me1 and H3K4me3) than for repressive marks (H3K27me3 and H3K9me3) and that, except for Signac, the range of appropriate bin size is relatively large (e.g., 50 kbp–1000 kbp for H3K27me3 or 10 kbp–200 kbp for H3K4me1). Furthermore, except for Signac, that range is relatively stable across methods for each bin size. We investigate in more details the reason why Signac behaves so distinctively in the supplementary text (Additional file 1: Fig. S1) and

show in particular that the fact that it uses a whitening step and a relatively high embedding dimension by default makes it might capture more noise for larger bin sizes.

We observe that using the GeneTSS annotation is usually not competitive compared to using an appropriate bin size. The fact that H3K4me3 is an exception to that rule is consistent with the fact that this mark is known to be particularly enriched around genes and TSSs. We can also see in Additional file 1: Fig. S5 that the various pseudobulk annotations, with either MACS2 or SICER, are also generally not competitive, with a less pronounced effect for H3K4me1 and H3K4me3. This is consistent with the fact that these marks tend to have small peaks, which are easier to identify with peak calling algorithms than larger ones.

It is interesting to note that the range of appropriate bin sizes for optimal representations usually includes 100kbp and can even go up to 500 kbp, which would a priori be considered too large to keep biological relevant information. In particular in [8], the authors made the choice of 5 kbp for H3K4me3 and 50 kbp for H3K27me3, while in [9], the authors chose 5 kbp for all marks, except for H3k4me3 for which it was 1 kbp. Here we find that, to reach a maximal concordance between epigenomic and transcriptomic embeddings, bin sizes one or two orders of magnitude larger than the ones used in previous studies are still competitive. This is likely due in part to the fact that the coverage per cell is so low that taking smaller bins introduces too much noise in the matrix, and to the fact that genes are not randomly distributed in the genome, and tend to cluster into groups of co-expressed genes [34, 35],. We can also observe from Fig. 3 that LSI based methods such as Signac can achieve good performances in the lower bin sizes regime (as well as ChromSCape_LSI as shown in the supplementary text).

### Selecting high coverage cells has a modest positive impact on the representation

In a standard QC pipeline, poorly covered cells can be filtered out before performing dimensionality reduction and subsequent analysis on the highest quality cells. Such selection step often leads to a trade-off between keeping a high number of cells to maximize the discovery rate of rare cell states and keeping only highly-covered cells to maximize the quality of the embedding. We now assess how selection of cells based on coverage affects the quality of the embedding, by applying different thresholds for coverage selection and measuring neighbor scores across methods.

As shown on Fig. 4, there is overall a modest gain in performance when applying more stringent QC criteria on cell coverage. Across histone marks, we observe a maximum gain of 15% and 13% in performance for H3K4me1 when using the best performing methods ChromSCape_LSI or Signac respectively (Additional file 1: Table S8). Across methods, we observe that the highest gains in performances are observed for the low performing methods identified above; see Additional file 1: Table S9. ChromSCape_PCA and SCALE benefit from a 41 and 21% gain respectively whereas ChromSCape_LSI only benefits from an average 8% gain. In summary, filtering out cells with low coverage has little impact on the quality of the representation, while reducing the probability to capture rare cells.

A related question important to prepare the experiments is, independently of the number of cells filtered out, to clarify the impact of average cell coverage. As studied in supplementary text, we observe that for a given number of cells, the average coverage

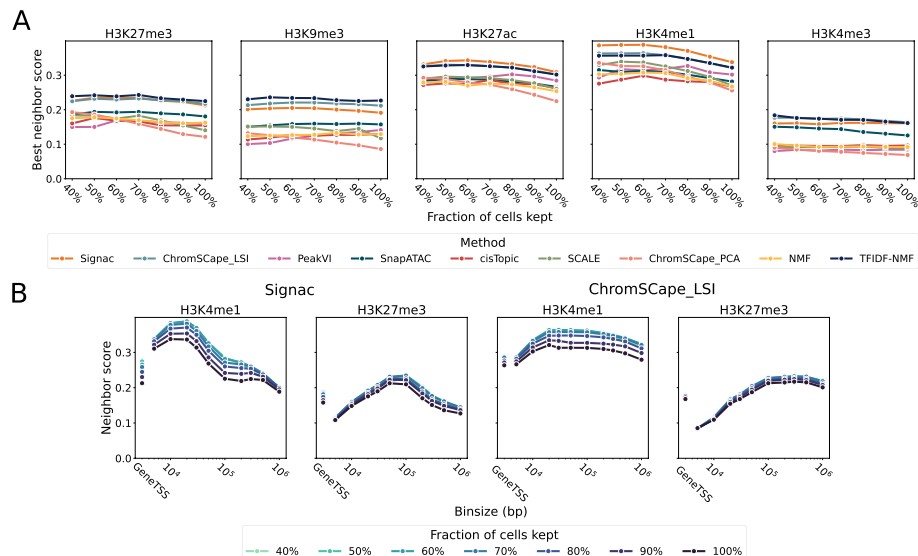

**Fig. 4 A** Each point corresponds to the best performance across matrix construction of a given method and a given coverage threshold, for the 7 methods, 5 marks, and 7 coverage conditions. **B** Performances of Signac and ChromSCape_LSI as a function of matrix construction on H3K4me1 and H3K27me3 for different coverage thresholds

per cell is strongly positively correlated with the quality of the representation for most marks and methods. This confirms that low cell coverage is currently one of the main reasons behind the difficulty to analyze scHTPM data and to capture a robust biological representation of each cell.

### Feature selection decreases the quality of the embedding

Another QC criteria used in single-cell analysis is the selection of features——genomic regions for single-cell epigenomics datasets——prior to dimensionality reduction. Two standard approaches are (i) the selection of regions with the highest coverage or (ii) the selection of regions that have a highly variable enrichment score across cells. Such a selection step is relatively common, but there is currently no consensus for scHPTM analysis on whether such selection is beneficial and which of the two methods is optimal.

To address this question, we compare the maximal neighborhood scores for all methods with various feature selection thresholds, when we select features based on variability (HVG) or coverage. The results are shown on Fig. 5. A and Additional file 1: Fig. S11 respectively, for the mouse brain dataset. We observe consistently that feature selection is generally detrimental to the performances, in the sense that for both methods, the more regions we keep the better the performances are. As shown on Fig. 5B for Signac and ChromSCape_LSI, this trend is in fact not only true when we look at the best performance reached over different bin sizes in the matrix construction step but also when we look at each bin size individually.

Feature selection has been shown to increase performances for scRNA-seq in [13] and is part of the guidelines for scATAC-seq [25]. Our results show that, contrary to scRNA-seq and scATAC-seq, feature selection is detrimental to the analysis of scHPTM data, and we therefore recommend not to use it.

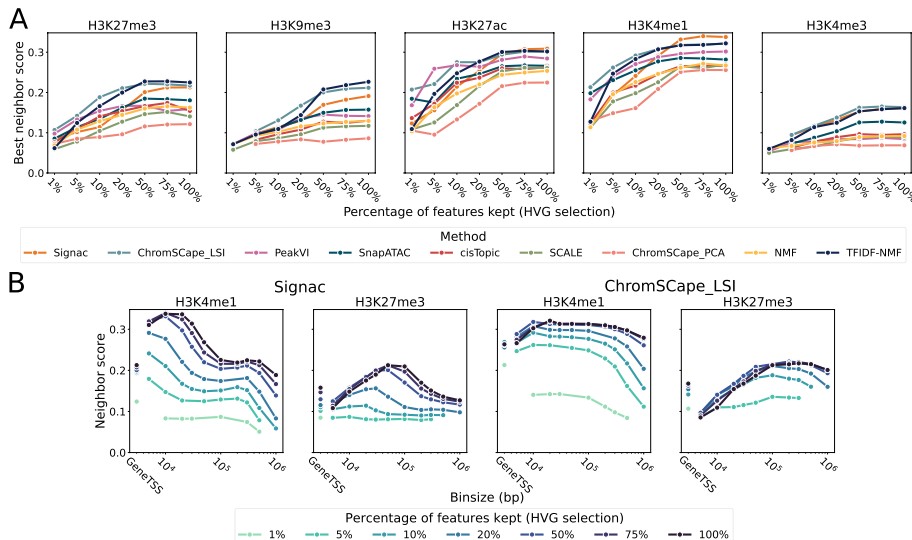

**Fig. 5** Role of feature selection, using the Highly Variable Gene (HVG) method used for scRNA-seq on the mouse brain dataset. **A** Each point corresponds to the best performance across matrix construction of a given method and a given percentage of features kept, for the 7 methods, 5 marks, and 7 feature selection conditions. **B** Performances of Signac and ChromSCape_LSI as a function of matrix construction on H3K4me1 and H3K27me3 for different feature selection thresholds

## Performances can reach a plateau with sufficient cell numbers

While computational parameters can have an important role in the quality of the representation [13], experimental ones also have a strong influence. In this section, we look at the role of the number of cells on such representations, in order to help practitioners design their experiments. For that purpose, we systematically downsample each dataset by randomly selecting a subset of cells of various size, and assess the quality of the representation obtained from the downsampled datasets. We show on Fig. 6A the best performance reached across matrix construction for each method on each mark, as a function of the size of the downsampled dataset, for the mouse brain dataset. We further add a finer grained sweep over dataset size for ChromSCape_LSI, by increasing the size of the datset by 500 cells per step as can be seen in Fig. 6B.

We see that all methods, on all datasets, benefit from an increase in the number of cells. However, it is interesting to note that the benefits resulting from a larger number of cells diminish as the number of cells increases. Indeed, we can observe that the performances increase quickly up to ~6000 cells and then only keep increasing at a much smaller rate. PeakVI is an exception to that observation, and we can see that its performances have not yet reached this plateau; see Additional file 1: Table S11. This is consistent with the intuition that deep learning based models require a large amount of data to achieve their full performances, and in the datasets used in this paper, this full performance does not seem to have been achieved. The gains in performances are also quite important, with an average increase of 34% by increasing the number of cells by 150% and 18% by increasing the number of cells by 66%.

On the other hand, the more standard methods, such as LSI or kernel PCA, reach their peak performances around 6000 cells, and only gain an average of 5% in performances by going from 6000 to 10,000 cells. Since these methods are the best performing

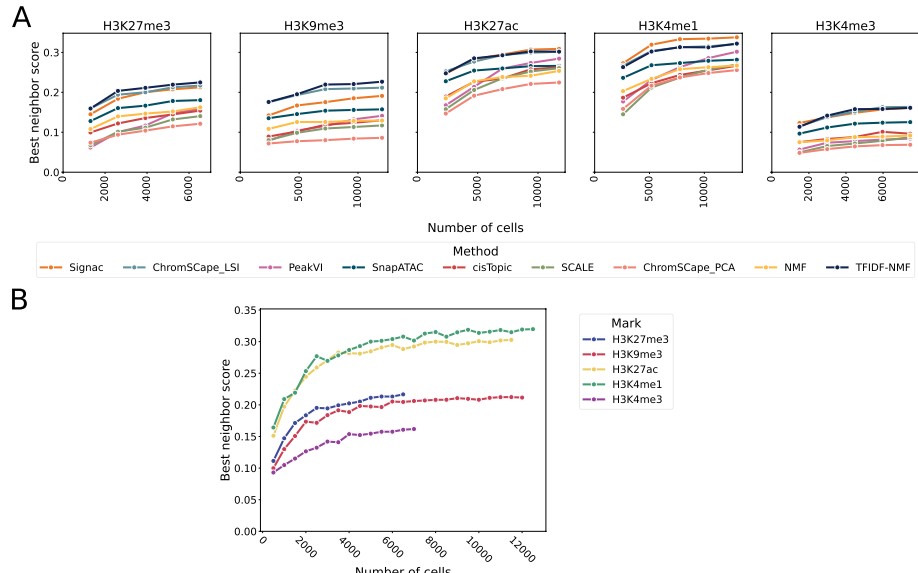

**Fig. 6** Effect of downsampling uniformly at random the number of cells in the experiment. Each point corresponds to the best performance across matrix construction. **A** Performances the 7 methods, on the 5 marks of the mouse brain dataset and on 5 sizes of dataset (by increase of 20% of the dataset size). **B** Performances of ChromSCape_LSI on the 5 marks, using an increase of 500 cells per step

ones in regime tested in this paper (less than 12,000 cells), it means that practitioners can sequence less cells while keeping relatively good performances. The case of Chrom-SCape_LSI is shown in more details in Fig. 6B, where the diminishing return effect of adding more cells is very pronounced. It also allows us to confirm that the difference in performances between the enhancing and repressive marks is not due to the number of cells present in the datasets, as we see a clear separation of 3 groups: H3K27ac and H3K4me1 having the best performances, H3K9me3 and H3K27me3 following them, and finally H3K4me3 having the worst performances. The plateauing effect can also be seen for all of these marks, leading us to believe that it is not specific to just some marks.

It is however very possible that given more cells (> 12,000), PeakVI and SCALE could outperform these methods and lead to better representations. This would be consistent with the behavior of deep learning-based methods on other modalities such as text of images, even though we can only conjecture that this would happen.

While the increase in the number of cells leads to observable and consistent gains in the quality of the representation, it is noteworthy that these gains have a lower influence than the use of an optimal matrix construction algorithm. It is also important to note that the performances of the current best methods do not strongly benefit from such an increase in the number of cells as can be seen in Additional file 1: Table S10, meaning that practitioners may work on relatively small samples while maintaining state-of-the-art performances.

## Discussion

In this paper, we studied the role of experimental parameters, cell and feature selection, matrix construction, and dimension reduction on the quality of the representation from scHPTM datasets. We decided to focus on the quality of the dimension reduction as it is

generally the input of most downstream tasks such as clustering, cell type identification, differential enrichment, or trajectory inference. A good representation is thus beneficial to all these tasks.

Unlike other benchmarks [12–14], we decided not to measure the quality of the representation based only on the ability of clustering algorithms to retrieve known cell types, because there are limited number of scHPTM datasets with high-quality labels. We explored the possibility of using the labels derived from co-assays, such as RNA or protein measurements. This approach allows us to be independent of labels, as well as working with potentially continuous cell states. Yet, when using this score, we make the assumption that cells that have similar epigenomic landscapes, measured by scHPTM, should at least locally display similar RNA or protein expression patterns. While we know that this assumption holds well for enhancer marks (e.g., H3K4me1 or H3K27ac), it might suffer some exceptions for repressive marks (H3K27me3, H3K9me3). We can note that this approach has already been successfully used in evaluating scATAC-seq pipelines in [18, 19]. In addition to this neighbor score, we have also included in our study measurements of ARI and AMI based solely on labels from authors, obtained computationally from the epigenomic datasets. In this case, the main limitation is that the method evaluation will tend to favor the method initially used by authors.

While we expected the choice of matrix construction algorithm to have an impact, that impact is larger than what we expected a priori. Indeed, as is shown in Additional file 1: Table S4, the performances using the best bin size can be up to 80% better than using the worst one. Surprisingly, the ranges of bin size are larger than what could be expected, and we were also surprised to find that enhancer marks such as H3K4me1 benefited from large bin sizes (up to 200 kbp) despite being known to accumulate into small peaks (in the order of a few kbps [17]). Yet, at a bin size of several 100 kbp, embeddings will not rely on local epigenomic enrichments, such as the ones observed for enhancers. The coverage of current scHPTM datasets might not be sufficient to produce reliable embeddings from small bins and may thereby be unable to distinguish cell states that differ by only few local enrichments. Identifying differences in enrichment for smaller regions than the bin size used for the embeddings can however be done when running differential enrichment analysis with a more appropriate bin size, while using the clusters obtained from the embedding for annotation. As our evaluation relies on the similarities between gene expression and gene regulation, it may be biased by the existence of large co-expressed gene clusters throughout the genome [34, 35]. With bin sizes over 100 kbp, we might be robustly detecting such co-regulated gene clusters. The fact that GeneTSS and pseudobulk-derived annotations were in general not competitive is also not something that was not previously rigorously established in the literature to the best of our knowledge.

It was also interesting to note that, except for PeakVI, the performances of the different methods tend to stagnate when increasing the number of cells. This is likely due to the relatively low complexity of the models used. More complex models such as PeakVI or SCALE did not manage to outperform these low complexity ones in our experiments. One could imagine that these models could show better performances with larger datasets, such as cell atlases, but they do not seem appropriate for experiments as they are currently designed.

On the other hand, we found that the performances of all methods largely benefited from being run on high coverage cells and that these performances did not stagnate on the available data, suggesting that future improvements of protocols——increasing coverage, such as in [36]——will surely provide additional information and granularity to refine embeddings.

We were also surprised to observe that feature selection, using either a variance or a coverage criteria, almost always had a negative impact on the performances. This may be due to the excessively low coverage per cells compared to other protocols where this procedure can be beneficial (see [13] for scRNA-seq).

To the best of our knowledge, this manuscript provides a comprehensive study on how to both design the experiment, build the matrix, and analyze scHPTM data. We hope that the large effect of matrix construction that we were able to identify will lead the community to pay more attention to this crucial, if overlooked step. Furthermore, by testing the algorithms and pipelines most likely to work on scHPTM data, we hope to save the community some time by avoiding having to discover which already existing algorithms work best.

## Conclusions

In conclusion, we propose the following best practices to start analyzing scHPTM data:

- Start by generating a count matrix with fixed bin sized for initial embedding (starting with 200 kbps for H3K9me3 and H3K27me3 and 100 kbp for H3K4me1, H3K4me3, and H3K27Ac, which best match cell identity). Finer resolution analysis, with smaller bins, can be performed, with the limitation that coverage per bin per cell will be lower.
- Avoid feature selection.
- Filter only barcodes not associated with a cell, and avoid filtering afterwards.
- Transform the matrix using TF-IDF.
- Use a matrix factorization algorithm on this transformed matrix (either SVD or NMF).

## Methods

### Matrix construction

We downloaded the mouse brain dataset from GSE152020. The data come in count matrix format, with 5 kbp bins for all marks except for H3K4me3 which is in 1 kbp bins. The larger bin sizes were obtained by merging the original bins together to form new bins using a custom script, available at https://github.com/vallotlab/benchmark_scepi genomics. The GeneTSS annotation comes from the ChromSCape package, and the matrix was done by merging the bins containing the regions in that annotation using the same custom script. We keep all the cells present in that matrix, as the original authors already applied QC steps on the cells.

The human PBMC dataset data was downloaded from https://zenodo.org/record/ 5504061; the data was processed from the fragment files. We used ChromSCape for generating 5 kbp matrices and then used our custom script to generate the other matrices

similarly to the mouse brain dataset. The MacsPseudoBulk annotation was obtained by turning the fragment files into bams, calling the peaks using MACS2, and then merging them using bedtools. The various SICER2 annotations were also generated by turning the fragment filed into beds, calling the peaks with SICER2 with island width of 200, 500, 2000, 5000, 20000, and a gap size thrice the size of the island size. We then select only the cells used in the original paper analysis, by keeping only the barcodes present in the rds objects on Zenodo.

The scChIP data was downloaded from GEO under the SuperSeries GSE164716, the matrices were constructed using ChromScape.

### In silico modifications

Using the matrices generated in the previous section, we modified them in order to both simulate experimental conditions, as well as apply standard bioinformatics steps. Feature selection was done using Seurat's `FindVariableFeatures` for the HVG selection and ChromSCape's `find_top_features` for the top regions selection, ran using our `filter_sce.R` script. For selecting only the high coverage cells, we sorted cells by coverage and kept only the most covered ones; the relevant script is `filter_cells_quality.R`. For studying the role of the number of cells, we sampled cells at ramdom without replacement from the matrice; the relevant script is `sample_cells.R`.

### scRNA-seq and CITE-seq processing

The scRNA-seq matrix for the mouse brain dataset was processed using the scanpy [21] package and following their best practice notebook (https://scanpy-tutorials.readthedocs.io/en/latest/pbmc3k.html). We have previously shown in [13] that the algorithms used in that package are robust and perform well. The CITE-seq matrix for the human PBMC dataset was extracted from the rds objects and processed with standard PCA.

### Representations for scHPTM

For computing the representations using the different methods, we used the implementation from the original packages, except for SnapATAC for which we used the reimplementation of [12] as it allowed a nicer API for running a large number of jobs; their implementation can be found on their github https://github.com/pinellolab/scATAC-benchmarking. For cisTopic, we ran the `runWarpLDAModels` method from the cisTopic Bioconductor package (version 0.3.0) and followed the steps from [12]. For Signac, we followed the scATAC-seq best practices vignette (https://satijalab.org/signac/articles/pbmc_vignette.html) and used the Signac CRAN package (version 1.7.0). For ChromSCape_LSI and ChromSCape_PCA, we processed the matrix with the `tpm_norm` and `TFIDF` methods respectively, then called the `pca` method, and removed the first principal component; all the methods were calllled from the ChromSCape Bioconductor package (version 1.6.0). For PeakVI, we followed the tutorial on the package website https://docs.scvi-tools.org/en/0.15.1/tutorials/notebooks/PeakVI.html using the scvi-tools (version 0.15.1) [37] pip package. Since SCALE did not have an API for calling their model, we modified the `main.py` script from the `scale` python package (version 1.1.0), so that it does not remove cells.

The scripts for processing used for all R methods are in the `R_analysis.R` script, PeakVI, and SCALE are respectively `peakVI_process.py` and `scale_process.py` scripts.

The R methods were run on CPUs with 16 cores and 32 GB of RAM, the deep learning ones (PeakVI and SCALE) on V100 GPUs with an additional 2 cores CPU.

### Neighbor score computation

To compute the neighbor score of an scHPTM representation, we first compute the $k$ nearest neighbor graphs (kNNG) for values of $k$ ranging from 0.1% up to 10% of the cells present in the dataset. We then compute the representation for the second modality using scanpy [21], whose algorithm (PCA) has been identified in [13] as being the most reliable for achieving good representations for scRNA-seq. We then compute kNNG on this second representation, count the number of common neighbors in the kNNG for each cell, divide by $k$, and average over the cells. This gives a score between 0 and 1, where 1 means that the two representations perfectly agree on which cells are similar, and a score of 0 means complete disagreement. We further average that score across the various values of $k$, which were selected to be 0.1%, 0.3%, 0.5%, 1%, 3%, 5%, and 10% of the cells contained in the assay, in order to take into account the multiple possible levels of similarity. Two completely random representations would have a score of 0.05 given the values of $k$ that we selected.

### Compute time

The runtime for each method has been measured on the H3K4me1 Mouse brain data, with a matrix built with 100 kbps. We measure the runtime as a function of the number of cells in Fig. 7 and as a function of the number of features in Fig. 8.

All measures were run on a computer with CPU with 32 GB of DDR4 RAM and an NVidia 2080Ti GPU. SCALE and PeakVI were trained using GPU acceleration. The time measured only accounts for training the models and generating the cells representation, reading the data and transforming it into the appropriate format was left out in order to avoid unfairly advantaging some methods over others.

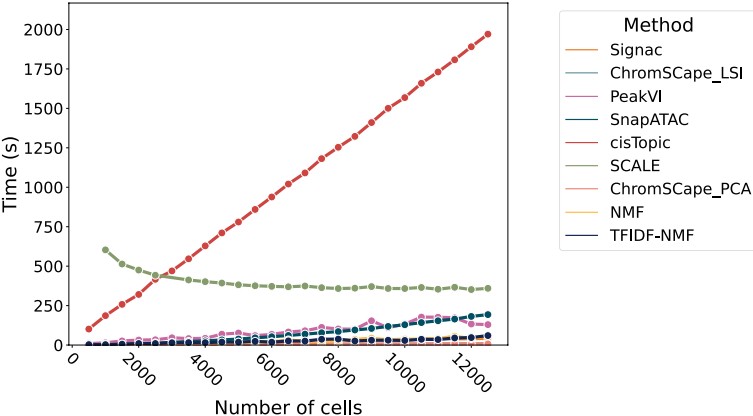

**Fig. 7** Runtime of the 9 dimension reduction methods as a function of the number of cells. The data used here is the Mouse Brain H3K4me1 with binsizes of 100 kbps and 26,360 features

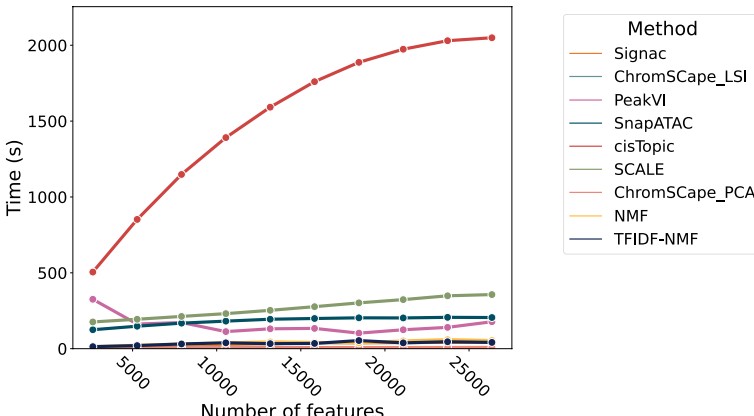

**Fig. 8** Runtime of the 9 dimension reduction methods as a function of the number of features. The data used here is the Mouse Brain H3K4me1 with 12962 cells

The fact that SCALE appears faster as the number of cells increases is due to the fact that it is trained by default by seeing a fixed number of cells (30000) instead of seeing all the cells multiple times. This should result in performances that are independent of the number of cells in the dataset, as reported in their paper [28]; however, it has to redo some computations each time it sees all the dataset; thus, having larger datasets reduces the number of times this computations have to be done. This training procedure is rather uncommon in the literature, but we chose to use the software as its authors recommended.

## Supplementary Information

---

**Additional file 1.** Contains the supplementary figures, tables, and text.

**Additional file 2.** Review history.

---

**Review history**
The review history is available as additional file 2.

**Peer review information**

**Authors' contributions**
FR, JPV, and CV designed the study and wrote the manuscript. PP performed the data formatting. FR performed the analysis.

**Funding**
Not applicable.

**Availability of data and materials**
The code used to perform the experiments and the analysis can be found on the github || have been deposited on Zenodo [38, 39].
 The mouse brain data was obtained from the NCBI Gene Expression Omnibus (GEO) (http://www.ncbi.nlm.nih.gov/geo/) under accession number GSE152020.
 The human PBMC data was obtained from Zenodo [40].
 The MDA-MB468 data was obtained from GEO, under accession number GSE164716.

## Declarations

### Ethics approval and consent to participate
Not applicable.

### Competing interests
The authors declare that they have no competing interests.

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

## 

