## [**Additional file 2.** Review history. · Genome Biology]

Review History

First round of review

Reviewer 1

Were you able to assess all statistics in the manuscript, including the appropriateness of statistical tests used? Yes, and I have assessed the statistics in my report.

Comments to author:

In this manuscript, Raimundo et. al., tried to identify several optimized parameters (including coverage and number of cells, of the count matrix construction method, of feature selection and normalization, and of the dimension reduction algorithm used that provide better representation of single-cell HPTM data.

The main results included 1) using fixed-size bin counts outperforms annotation-based binning; 2) dimension reduction methods based on latent semantic indexing outperform others; 3) feature selection is detrimental, while keeping only high-quality cells has little influence on the final representation if enough cells are analyzed. I agree that a systematical study for finding the best representation of data is important, but conclusions should be carefully and rigorously drawn, because the conclusions might be used as benchmarks by other researchers. Overall, I think the study and conclusions in this current manuscript are not solid enough. Below are my specific comments:

1) The authors should avoid using the term "best practices" in the title since it is misleading. The authors just considered a small subset of methods and parameters. Also, current study just used a specific set of data. However, single-cell histone data generated from different methods such as Tn5-based or MNase-based can have different optimized parameters. The authors should use a much more specific title to present exactly what they study to avoid confusion.

2) It is difficult to judge the comparisons. First, colors for labeling in figure 2B and other similar figures were repeatedly used. Thus, there is no way to judge how good the clustering is based on the figures. Second, the author explained the use of neighbor score is supported by Ref.19 and other refs. However, neighbor score is just one of the measurements which cannot fully reflect the high-dimensional comparison. To overcome this, the study from ref 19 used several metrics including clustering-based metrics, Cell type average silhouette width, label score, and neighbor score. The authors should also use all these metrics for making a robust and valid conclusion.

3) It is not convincing that using fixed-size bin counts outperforms annotation-based binning because the authors just considered a small subset of annotation-based binning methods. In particular, the authors used MACS to call peaks from pseudo-bulk data. Peaks called from MACS are based on normalized density. However, log2 based transform is usually applied to the density values. Therefore, the authors should also try other peak calling methods that are

based on log₂ density such as SICER. Next, another considering factor is whether to use the width reported by SICER or to use a fixed width. If a fixed width is choosing, what is the best fixed width? I think the authors should consider all these factors.

4) The authors considered seven methods for analyzing the count matrices: cisTopic, Signac SnapATAC, PeakVI, SCALE, and ChromSCape with TF-IDF (ChromSCape_LSI) and count per million (CPM) normalization (ChromSCape_PCA). That these methods are popular does not necessarily mean that they are the best candidates. Other studies also used other methods like Non-negative matrix factorization, Laplacian Matrix, False Nearest Neighbors, etc. If the authors just focus on the seven methods, then the authors should state explicitly and clearly that the study is to find choice among the seven methods.

Reviewer 2

Were you able to assess all statistics in the manuscript, including the appropriateness of statistical tests used? No, I do not feel adequately qualified to assess the statistics.

Comments to author:

Raimundo et al. performed a computational benchmark to assess the impact of experimental parameters and of the data analysis pipeline on the ability of the cell representation produced to recapitulate known biological similarities from scCUT&Tag. They systematically studied the impact of coverage and number of cells, of the count matrix construction method, of feature selection and normalization, and of the dimension reduction algorithm used. The authors also studied the data from different species, including human and mouse. Overall, analysis of the benchmark results is well described, and the paper is timely for the single-cell histone post translation modification assays computational guidance. However, the manuscript can be further enhanced by fixing the following comments.

Major comments:

1. The authors analyzed the data from co-assays and used the neighbor score to assess the performances. While it was fine when they used scRNA-seq and CITE-seq as reference, it may ignore the possibility that these two omics layers are asynchronous. It would be interesting to see how the parameters affect other metrics from scCUT&Tag data itself, such as ARI and AMI. In addition, the authors only used two datasets, and they should include more.
2. The authors only studied the data from scCUT&Tag, so they should clarify this upfront. Otherwise, they should also include other types of data, such as scChIP-seq.
3. The authors should share their code, which is essential for reproducing their data analysis.
4. For the p-values of pairwise comparisons (e.g. Table S7), are they multiple testing corrected?
5. Page 8 line 47, what is the QC criteria was used to select cells?

6. The authors found that feature selection will decrease the quality of the embedding, but how it will affect the computation time when we use more features? Similar to the number of cells, how it will affect the computation time when we use more cells. Please also list the computer configuration when they report the computation time.

7. The authors should add a list of the best practices for selecting appropriate parameters, either using a table or figure.

Minor comments:

1. Page 2 line 37, "10.000" should be "10,000"?

2. Page 6 line 53, I didn't see Fig. S9

3. The x axis label of Figure 3A is somewhat confusing, "Binsize (bp)" can be moved toward right side.

4. Page 8 line 6, I didn't see the results from Seurat.

We thank the reviewers and the editor for their relevant and helpful comments. We describe below the changes we made in the revision, and provide a detailed answer to each question raised by the reviewers.

Reviewer #1:

In this manuscript, Raimundo et. al., tried to identify several optimized parameters (including coverage and number of cells, of the count matrix construction method, of feature selection and normalization, and of the dimension reduction algorithm used that provide better representation of single-cell HPTM data.

The main results included 1) using fixed-size bin counts outperforms annotation-based binning; 2) dimension reduction methods based on latent semantic indexing outperform others; 3) feature selection is detrimental, while keeping only high-quality cells has little influence on the final representation if enough cells are analyzed. I agree that a systematical study for finding the best representation of data is important, but conclusions should be carefully and rigorously drawn, because the conclusions might be used as benchmarks by other researchers. Overall, I think the study and conclusions in this current manuscript are not solid enough. Below are my specific comments:

1) The authors should avoid using the term “best practices” in the title since it is misleading. The authors just considered a small subset of methods and parameters. Also, current study just used a specific set of data. However, single-cell histone data generated from different methods such as Tn5- based or MNase-based can have different optimized parameters. The authors should use a much more specific title to present exactly what they study to avoid confusion.

We agree with the reviewer that the term ‘best practices’ was not adapted. As we have now included datasets from different methods, we have changed the title to ‘A benchmark of computational pipelines for single-cell histone modification data’.

Following the reviewer’s comment, we now combine data from two different methods (Tn5 and MNase-based). We have added experiments using an MNase-based protocol (scChIP-seq), taken from Marsolier et al. where the cells come from two different experimental conditions (untreated vs treated MDA-MB468 breast cancer cells). This dataset, containing experimental labels, was evaluated with ARI/AMI.

We can observe (Figure 1R1) the three top performing methods are TFIDF-NMF, NMF and PeakVI, followed by ChromSCape_LSI and cisTopic. Signac suffers from the fact that it relies on 50 dimensions for its embedding, which may introduce noise as the dataset is not complex. We can see on Fig 1R2B, that the increase in binsize up to 200kbp is beneficial (ARI for TFIDF-NMF).

Fig 1R1: Study of scChIP-seq dataset. **A.** Best performances achieved by each method across all matrix constructions. **B.** UMAP projection at various binsizes for TFIDF-NMF coloured by treatment status.

2) It is difficult to judge the comparisons. First, colors for labeling in figure 2B and other similar figures were repeatedly used. Thus, there is no way to judge how good the clustering is based on the figures. Second, the author explained the use of neighbor score is supported by Ref.19 and other refs. However, neighbor score is just one of the measurements which cannot fully reflect the high-dimensional comparison. To overcome this, the study from ref 19 used several metrics including clustering-based metrics, Cell type average silhouette width, label score, and neighbor score. The authors should also use all these metrics for making a robust and valid conclusion.

Regarding color issues, we have now adapted colors in the figures, using different colors for the methods, marks, and cell types. Following the reviewer's suggestion, we have now also added supervised metrics - AMI/ARI (with both hierarchical and K-means clustering) - to our analysis using either the co-assay to compute labels for cells, or experimental conditions for the two new datasets we added. Following a subsequent suggestion (point 4 below), we have also added an additional method, NMF with or without TF-IDF. Altogether, using these metrics, we observe like before that TF-IDF based methods recurrently outperform others, with the exception of PeakVI that becomes the second best performing method on H3K4me1 and H3K27ac in the mouse brain data, followed by cisTopic, SCALE and SnapATAC (Figure 1R2-R3).

The performances evaluated using AMI or ARI are almost exactly the same, as seen in Fig 1R4, and so are the performances using K-means or hierarchical clustering (Fig 1R4-R5), so we chose to only report K-means ARI in the manuscript for the sake of conciseness.

We can also observe that the effect of the matrix construction is less pronounced using these supervised metrics, which we believe is due to the quality of the computational labels on the co-assay. We have included these results in Figure S8-S11 of the manuscript.

Fig 1R2: ARI of the 9 dimension reduction algorithms on the 5 marks in the mouse brain dataset. **A.** Best performances across the various matrix construction methods. **B.** Performances as a function of the matrix construction.

Fig 1R3: ARI of the 9 dimension reduction algorithms on the 5 marks in the human PBMC dataset. **A.** Best performances across the various matrix construction methods. **B.** Performances as a function of the matrix construction.

Fig 1R4: Joint distribution of the AMI and ARI using k-means clustering on the various datasets.

Fig 1R5: Joint distribution of the ARI, using either hierarchical clustering with ward's linkage or k-means, on the various datasets.

3) It is not convincing that using fixed-size bin counts outperforms annotation-based binning because the authors just considered a small subset of annotation-based binning methods. In particular, the authors used MACS to call peaks from pseudo-bulk data. Peaks called from MACS are based on normalized density. However, log₂ based transform is usually applied to the density values. Therefore, the authors should also try other peak calling methods that are based on log₂ density such as SICER. Next, another considering factor is whether to use the width reported by SICER or to use a fixed width. If a fixed width is choosing, what is the best fixed width? I think the authors should consider all these factors.

We agree with the reviewer that using MACS might have been limited to assess annotation-based binning. We have now also used SICER with width of [200, 500, 2000, 5000, 20000 bp]. We have observed that while the quality of the representations are affected by the use of SICER instead of MACS, there is no consistent effect across method or mark (Figure 1R6). For each mark/method, we can still identify a fixed bin size that outperforms the use of a peak caller on pseudo-bulk, whatever the size of the peak. We have now added this analysis to our manuscript in Figure S5.

Figure 1R6 (Updated Fig S5): Neighbor score for the 9 dimension reduction algorithms on the 5 marks in the human PBMC dataset, as a function of the matrix construction. The different values in SicerPseudoBulk correspond to different width parameters for the islands, the values in Bins correspond to the various binsizes used.

4) The authors considered seven methods for analyzing the count matrices: cisTopic, Signac, SnapATAC, PeakVI, SCALE, and ChromSCape with TF-IDF (ChromSCape_LSI) and count per million (CPM) normalization (ChromSCape_PCA). That these methods are popular does not necessarily mean that they are the best candidates. Other studies also used other methods like Non-negative matrix factorization, Laplacian Matrix, False Nearest Neighbors, etc. If the authors just focus on the seven methods, then the authors should state explicitly and clearly that the study is to find choice among the seven methods.

We had initially selected methods based on the facts that (i) the method was publicly available on github, (ii) the method was used by at least one publication with single-cell histone or ATAC-seq datasets, (iii) was identified as a high performer on scATAC-seq data. We had missed the use of NMF for scATAC-seq (e.g. scOpen) and have now added it to our benchmark, with and without TF-IDF transformation (TFIDF-NMF and NMF respectively). To the best of our knowledge, the last two methods (Laplacian Matrix & False Nearest Neighbors) have not yet been used for the analysis of single-cell data.

Thanks to this addition we found that TFIDF-NMF is extremely competitive, further supporting the observation that TF-IDF transformation is key to obtaining meaningful embeddings and that LSI based methods (understood as TF-IDF followed by matrix factorization) are the most competitive. The effects of feature selection, sample selection, number of cells, and coverage are the same for these two methods as for the other 7.

We have updated all figures to include the two new methods, and show updated Fig 2-3 here.

Updated Figure 2: A. Best performances of the different representation methods on the mouse brain dataset. **B.** UMAP representation of the different samples in the mouse brain dataset, the first row is the RNA co-assay processed with PCA using the scanpy best practices, the second row is the scHPTM assay processed with ChromScape_LSI using the matrix construction algorithm with the best neighbor score, both colored by the labels of the authors obtained from the scRNA-seq co-assays

Updated Figure 3: A. Neighbor score performances of the 9 dimension reduction algorithms on the 5 marks in the mouse brain dataset, as a function of the matrix construction. **B.** UMAP projection of H3K4me1 and H3K27me3 using ChromSCape_LSI using bins of 20kbp and 300kbp, colored by the labels of the authors obtained from the scRNA-seq co-assays.

Reviewer #2:

Raimundo et al. performed a computational benchmark to assess the impact of experimental parameters and of the data analysis pipeline on the ability of the cell representation produced to recapitulate known biological similarities from scCUT&Tag. They systematically studied the impact of coverage and number of cells, of the count matrix construction method, of feature selection and normalization, and of the dimension reduction algorithm used. The authors also studied the data from different species, including human and mouse. Overall, analysis of the benchmark results is well described, and the paper is timely for the single-cell histone post translation modification assays computational guidance. However, the manuscript can be further enhanced by fixing the following comments.

1. The authors analyzed the data from co-assays and used the neighbor score to assess the performances. While it was fine when they used scRNA-seq and CITE-seq as reference, it may ignore the possibility that these two omics layers are asynchronous. It would be interesting to see how the parameters affect other metrics from scCUT&Tag data itself, such as ARI and AMI. In addition, the authors only used two datasets, and they should include more.

We agree with the reviewer that cells could harbour conflicting epigenomic and transcriptomic characteristics - thinking about epigenomic priming of enhancers or promoters for example. As mentioned in our manuscript, the limitation to the independent evaluation of single-cell histone modification datasets, is the absence of labels. For all the datasets generated with scCUT&Tag that we could find, annotations were obtained computationally. In this case, we need to keep in mind that computing ARI/AMI will favour the initial pipelines used to obtain the labels.

We have added one dataset (from Marsolier et al., Nature Genetics 2022) where the labels are not computationally derived, and evaluated with ARI the different methods. The labels are the treated vs untreated status of MDA-MB468 cells (human breast cancer) receiving either DMSO (placebo) or 5FU (chemotherapy treatment). Using this dataset we computed informative rankings of the methods, with the best ones being TFIDF-NMF, NMF and PeakVI. This is presented in Fig 2R1.

Fig 2R1: A. Best performances achieved by each method across all matrix constructions. **B.** UMAP projection at various binsizes for TFIDF-NMF coloured by treatment status.

We have also added AMI and ARI for the scCUT&Tag datasets using the labels from the original papers; we have used k-means and hierarchical clustering with ward linkage and specified a number of clusters equal to the number of cell types. Performances using AMI or ARI are extremely correlated (Fig 2R2), same goes for using either Ward or k-means (Fig 2R3), so we chose to only include ARI using k-means in the manuscript for succinctness sake.

Using the supervised metrics, the relative rankings of the method are overall conserved, but the effect of matrix construction seems less pronounced. This may be due to the quality of the labels (which are computational in nature), or to the fact that clustering is a less sensible task than representation.

We have kept in the main figures the usage of the neighbor score, and added in supplementary figures other metrics (Fig S8-S9). Indeed, the neighbor score measures the quality of the embedding itself, which has effect on a variety of tasks, not limited to clustering, e.g trajectory inference.

Fig 2R2: Joint distribution of the AMI and ARI using k-means clustering on the various datasets.

Fig 2R3: Joint distribution of the ARI, using either hierarchical clustering with ward’s linkage or k-means, on the various datasets.

2. The authors only studied the data from scCUT&Tag, so they should clarify this upfront. Otherwise, they should also include other types of data, such as scChIP-seq.

As mentioned in the previous answer we have now added experiments using scChIP-seq.

3. The authors should share their code, which is essential for reproducing their data analysis.

We indeed forgot to add the analysis code to the repository, this has been fixed. We have also made a section: “code availability” instead of just having a link to our github in the “matrix construction” section.

4. For the p-values of pairwise comparisons (e.g. Table S7), are they multiple testing corrected?

We indeed did not correct initially for multiple testing. Using correction there are not enough samples for the corrected p-values to be significant - even if a method is superior to another in all cases. Adding more datasets is not an option as we did not find other datasets that satisfy our co-assay criterion. We have therefore removed the p-values from the text and stated when methods were superior in all cases to others.

5. Page 8 line 47, what is the QC criteria was used to select cells?

We used the number of reads per cell, we have now indicated that in the text.

6. The authors found that feature selection will decrease the quality of the embedding, but how it will affect the computation time when we use more features? Similar to the number of cells, how it will affect the computation time when we use more cells. Please also list the computer configuration when they report the computation time.

We have now analyzed runtime as a function of the number of cells and features (Fig 2R4-2R5). We observe that the slowest method is cisTopic, followed by the deep-learning based methods. Most methods follow a linear runtime of the number of samples and of features. The apparent decrease in runtime of SCALE is most probably due to its training scheme (being more efficient on larger epochs).

Figure 2R4: Runtime in seconds of the various methods depending on number of cells, on the mouse brain H3K27me3 matrix built with 100kpbs windows. This matrix was downsampled in cells in order to produce datasets with various numbers of cells in them.

Figure 2R5: Runtime in seconds of the various methods depending on number of features, on the mouse brain H3K27me3 matrix built with 100kpbs windows. This matrix was downsampled in features in order to produce datasets with various numbers of features in them.

7. The authors should add a list of the best practices for selecting appropriate parameters, either using a table or figure.

We have now added a list containing our recommended practices in a box at the end of the manuscript (Box 1).

Box 1:

- Start by generating a count matrix with fixed bin sized for initial embedding (starting 200kpbs for H3K9me3 and H3K27me3, and 100kbp for H3K4me1, H3K4me3 and H3K27Ac).
- Avoid feature selection
- Filter only barcodes not associated with a cell, and avoid filtering afterwards.
- Transform the matrix using TF-IDF.
- Use a matrix factorization algorithm on this transformed matrix (either SVD or NMF).

Minor comments:

1. Page 2 line 37, "10.000" should be "10,000"?

Fixed

2. Page 6 line 53, I didn't see Fig. S9

This was a typo on our end, this is now referred to as Table S6.

3. The x axis label of Figure 3A is somewhat confusing, "Binsize (bp)" can be moved toward right side.

Fixed

4. Page 8 line 6, I didn't see the results from Seurat.
Indeed, it was an oversight on our end, we meant Signac.
Fixed.

Second round of review

Reviewer 1: The authors have done a nice job in revising the manuscript. I have no more concerns.